# Regulation of Microglial Signaling by Lyn and SHIP-1 in the Steady-State Adult Mouse Brain

**DOI:** 10.3390/cells12192378

**Published:** 2023-09-28

**Authors:** Erskine Chu, Richelle Mychasiuk, Evelyn Tsantikos, April L. Raftery, Elan L’Estrange-Stranieri, Larissa K. Dill, Bridgette D. Semple, Margaret L. Hibbs

**Affiliations:** 1Department of Neuroscience, Central Clinical School, Monash University, Melbourne, VIC 3004, Australia; 2Department of Immunology, Central Clinical School, Monash University, Melbourne, VIC 3004, Australia; evelyn.tsantikos@monash.edu (E.T.); april.raftery1@monash.edu (A.L.R.); elan.lestrange-stranieri1@monash.edu (E.L.-S.); 3Department of Neurology, Alfred Health, Melbourne, VIC 3004, Australia; 4Department of Medicine, The Royal Melbourne Hospital, The University of Melbourne, Melbourne, VIC 3010, Australia

**Keywords:** immune responses, inflammation, immune cell signaling, microglia, neuroinflammation

## Abstract

Chronic neuroinflammation and glial activation are associated with the development of many neurodegenerative diseases and neuropsychological disorders. Recent evidence suggests that the protein tyrosine kinase Lyn and the lipid phosphatase SH2 domain-containing inositol 5′ phosphatase-1 (SHIP-1) regulate neuroimmunological responses, but their homeostatic roles remain unclear. The current study investigated the roles of Lyn and SHIP-1 in microglial responses in the steady-state adult mouse brain. Young adult Lyn−/− and SHIP-1−/− mice underwent a series of neurobehavior tests and postmortem brain analyses. The microglial phenotype and activation state were examined by immunofluorescence and flow cytometry, and neuroimmune responses were assessed using gene expression analysis. Lyn−/− mice had an unaltered behavioral phenotype, neuroimmune response, and microglial phenotype, while SHIP-1−/− mice demonstrated reduced explorative activity and exhibited microglia with elevated activation markers but reduced granularity. In addition, expression of several neuroinflammatory genes was increased in SHIP-1−/− mice. In response to LPS stimulation ex vivo, the microglia from both Lyn−/− and SHIP-1−/− showed evidence of hyper-activity with augmented TNF-α production. Together, these findings demonstrate that both Lyn and SHIP-1 have the propensity to control microglial responses, but only SHIP-1 regulates neuroinflammation and microglial activation in the steady-state adult brain, while Lyn activity appears dispensable for maintaining brain homeostasis.

## 1. Introduction

Neuroinflammation is a core defense mechanism of the central nervous system that is upregulated to remove noxious agents, clear bacteria or viruses, and promote healing responses [1]. Microglia and astrocytes are the primary innate immune cells responsible for neuroinflammation as they undergo biochemical and morphological changes and release inflammatory cytokines and chemokines when challenged. Although these processes are typically transient, aberrant or persistent activation can be detrimental to neuronal health by promoting synapse degeneration, metabolic dysfunction, glutamate dysregulation, and excitotoxicity [2]. As such, dysregulated neuroimmune responses have been increasingly implicated in various neurodegenerative diseases and psychological disorders [3,4].

The phosphoinositide-3 kinase (PI3K)-AKT signaling pathway is a highly conserved pathway that controls numerous facets of cellular function, including cell growth, survival, and responses to extracellular stimuli. In immune cells, it also regulates specific mechanisms such as cytokine release, cell migration, and antibody production [5]. Abnormal PI3K-AKT signaling has been associated with the development of peripheral inflammatory diseases in patients and in animal models [6]. The activity of this pathway is tightly regulated through stepwise activation and deactivation by kinases and phosphatases [7]. Src family non-receptor protein tyrosine kinases are well-known activators of PI3K signaling by phosphorylating cell surface receptors to facilitate the recruitment and activation of PI3K. In immune cells, one of the lipid phosphatases that negatively regulates PI3K signaling is the lipid phosphatase SHIP-1. Another negative regulator of immune cell signaling is the Src family kinase Lyn. Lyn activity is augmented by autophosphorylation [8], which then enables it to phosphorylate sites on inhibitory cell surface receptors to recruit SHIP-1, whereby Lyn directly phosphorylates and activates SHIP-1 [9,10]. Evidence of a critical role for these negative regulators comes from knockout mouse studies. Global ablation of Lyn or SHIP-1 in mice results in augmented activity of B lymphocytes and leads to the development of autoimmune and inflammatory lupus-like disease [11,12], as well as hyperresponsiveness to growth factors and infiltration of innate immune cells into systemic organs [10,13,14,15,16]. SHIP-1-deficient mice also exhibit chronic inflammatory lung disease, osteoporosis, and Crohn’s disease-like ileitis [12,16,17,18,19].

Lyn and SHIP-1 are also expressed in the central nervous system. Lyn is expressed widely in the brain, where it is reported to regulate downstream signaling of various neurotrophic factors [20,21,22,23,24]. It is also highly expressed in glial cells, including human and mouse microglia and oligodendrocytes [25]. The regulatory role of Lyn in glial cells is unclear, but it is believed to be partially involved in mediating microglial activation and promoting myelination and proliferation of oligodendrocytes [22,24,26,27]. Conversely, in the brain, SHIP-1 is predominantly found in microglia due to their myeloid origin [28,29], although its expression has also been reported in endothelial cells, albeit to a lesser extent [30].

Recent studies have linked Lyn and SHIP-1 activity to the development of neurodegenerative diseases [16,31]. For example, activation of Lyn was detected in the hippocampus of patients with Alzheimer’s disease [32]. Exposure of neuronal cells to Aβ_1-42_ peptides induced the activation of Lyn and led to phosphorylation of the inhibitory motif in the FcγRIIb receptor and neuronal cell death. Inhibition of this interaction with a small molecule Lyn inhibitor was found to prevent Aβ_1-42_-mediated neurotoxicity [32,33]. Moreover, activation of Lyn was reportedly increased in microglia from Alzheimer’s patients and in rodent models after exposure to Aβ_1-42_ oligomers [33,34,35]. Recent genome-wide association studies have identified single-nucleotide polymorphisms within the *INPP5D* (SHIP-1) gene as a risk factor for the development of Alzheimer’s disease [36,37,38,39,40]. It is speculated that these polymorphisms alter the activity of SHIP-1, which interferes with microglial uptake of debris or surrounding plaques, and lysosomal compartment formation [28,40].

Taken together, the literature suggests that Lyn and SHIP-1 may be involved in regulating microglial responses and contribute to the development of neurodegenerative disease. However, their high expression levels, even under homeostatic conditions, suggest that they may also have important functions in the steady-state brain. Therefore, this study aimed to examine whether Lyn and SHIP-1 have a regulatory role in the naïve adult brain.

## 2. Materials and Methods

### 2.1. Animals

Young adult mice of 10–12 weeks of age were used for all studies. The animals used were as follows: 1) male and female Lyn−/− (*Lyn^tm1Ard^*) mice [11] backcrossed to a C57BL/6 background [41], compared to control C57BL/6 mice; and 2) male and female SHIP-1−/− mice (*Inpp5d^tm1Dmt^*) [14] on a C57BL/6 background [12] compared to littermate SHIP-1+/+ mice for all experiments except for flow cytometry analysis, where littermate SHIP-1+/− mice were used as controls (SHIP-1+/+ and SHIP-1+/− presented throughout as control). This approach was chosen as previous studies have reported that SHIP-1+/− mice do not have a phenotype and are identical to SHIP-1+/+ mice [12,13], and we have now shown that they do not exhibit behavioral differences (Appendix A). The group numbers used in all experiments are described in Appendix A. Mice were housed in the specific pathogen-free facility in the Precinct Animal Centre at the Alfred Research Alliance under a 12 h light–dark cycle, with unrestricted access to food and water.

### 2.2. Behavior Testing

Behavior testing was conducted on mice at 10–11 weeks of age in the following order over a 1-week period. Anxiety-like behavior was assessed using the Elevated Plus Maze (EPM) (10 min duration), where the time that mice spent in the well-lit “open-arms” was measured, as previously described [42]. General locomotor activity was assessed using an open-field test (400 mm W × 400 mm D × 300 mm H) (10 min duration), with anxiety as a secondary measure based on time spent in the center area compared to the periphery of the arena [43]. Gross motor functioning and coordination were assessed using the accelerating rotarod test, averaged across 3 trials per day (with a 30 min inter-trial interval) over two consecutive days, with the rotarod accelerating from 4 to 40 rpm over a 5 min period [44,45]. Working and spatial memory was evaluated using the Y-maze (15 min habitation, 30 min inter-trial interval, followed by 5 min test phase) as previously described [42], and analyzed using the discrimination index ($Discrimination\;index=\frac{{Time}_{Novel}-{Time}_{Familiar}}{{Time}_{Novel}+{Time}_{Familiar}}$). Animal movement during EPM, open-field, and Y-maze was filmed using an overhead camera and automatically tracked using TopScan Lite software (v.2.00, Clever Sys Inc., Reston, VA, USA). All tests were performed between 8 am and 3 pm, as detailed previously [42], and analyses were performed by an investigator blinded to the experimental group.

### 2.3. Transcardial Perfusion and Tissue Fixation

Transcardial perfusion and tissue fixation procedures were performed as previously described [46]. Briefly, the mouse was terminally anesthetized with sodium pentobarbitone (80 mg/kg i.p.), and their spleens were extracted and weighed. The mouse underwent transcardial perfusion with saline (3 mL/min for 5 min) followed by 4% paraformaldehyde. The whole brain was collected, post-fixed in 4% paraformaldehyde overnight and then immersed in 30% sucrose for 3–5 days before being embedded in Optimal Cutting Temperature (OCT) media (ProSciTech, Kirwan, Australia) for sectioning.

### 2.4. Immunofluorescence

Perfused brains were cryo-sectioned as 12 µm coronal sections, and slices between Bregma 0.7 mm and 3.5 mm (Allen Mouse Brain Atlas; available from atlas.brain-map.org) were collected onto Superfrost Plus slides (Thermo Fisher Scientific, Waltham, MA, USA). Systematically sampled sections (e.g., every 6th section, for a total of 8 sections per brain) were incubated with normal donkey serum for 1 h to block non-specific binding. Tissue was then incubated with an antibody against ionized calcium binding adaptor molecule 1 (IBA-1) (goat polyclonal, AB5076, 1:500, Abcam, Cambridge, UK) at 4 °C overnight. Secondary donkey anti-goat Alexa Fluor 488 antibody (1:250, Invitrogen, Waltham, MA, USA) was applied and incubated for 1 h at room temperature. Tissue was then counterstained with Hoechst (1:1000, Sigma-Aldrich, St. Louis, MO, USA) and mounted with glass coverslips using fluorescence mounting media (Agilent, Santa Clara, CA, USA).

All fluorescence images were captured using a fluorescent Nikon-TiE inverted microscope and NIS-Elements software (5.21.00, Nikon, Tokyo, Japan), and were analyzed with ImageJ software (NIH ver. 1.52p, National Institutes of Health, Bethesda, MD, USA). IBA-1 immunofluorescence intensity was measured by binary conversion followed by manual thresholding, and expressed as the percentage of positive signal within each region of interest [42]. The quantification methods were uniformly applied to all tissue sections/images by an investigator blinded to the experimental groups.

### 2.5. Microglia Morphology Analysis

Analysis of microglial morphology was conducted as described previously [47]. In brief, five randomly selected IBA-1+ microglia were isolated from the three most medial images of the cortex of each brain, for a total of 15 microglia per brain, and 120 microglia per experimental group. Cell branches and soma were traced and skeletonized using ImageJ (NIH ver. 1.52p), and the number of branches, average branch length, and soma area were measured for each cell.

### 2.6. Cresyl Violet Staining

Coronal sections were stained with cresyl violet acetate (0.25% *w*/*v*, Sigma-Aldrich, St. Louis, MO, USA), then dehydrated with increasing ethanol concentrations, and cover-slipped using DPX mountant (Sigma-Aldrich). Images were captured using the Leica Aperio AT Turbo Brightfield slide scanner at 20× magnification (Monash Histology Platform, Monash University, Melbourne, Australia). The volume of key brain regions of interest (cortex and hippocampus) was estimated using the Cavalieri method of unbiased stereology as previously described using ImageJ (NIH ver. 1.52p) [42,48].

### 2.7. Flow Cytometry Analysis

Microglial phenotypes were assessed by flow cytometry analysis as previously described [46]. Briefly, whole saline-perfused brains were collected, along with spleens. Brains were homogenized using gentleMACS tissue dissociation (Miltenyi Biotec, Bergisch Gladbach, Germany) in digestion buffer containing RPMI medium (Thermo Fisher Scientific), 1% fetal calf serum (Thermo Fisher Scientific), 2.5% Liberase (Sigma-Aldrich), 1% HEPES (Thermo Fisher Scientific), and 1% DNAse I (Sigma-Aldrich) for 30 min at 37 °C. Digested brain tissue was filtered through a 70 μm nylon mesh and isolated single cells were resuspended in cold PBS. Debris Removal Solution (Miltenyi Biotec) was used according to the manufacturer’s instructions to remove fatty debris. Splenocytes were extruded from under the spleen capsule, washed, and filtered, and thereafter used for single-stain controls.

Single cell suspensions of brain and spleen were first incubated with rat anti-mouse CD16/CD32 (mouse Fc Block, produced in-house) for 20 min at 4 °C to block non-specific binding of antibodies. Cells were then incubated for 20 min at 4 °C with receptor-specific antibodies to identify microglial cells (CD11b, CD14, CD206, CD40, CD45, CD86, CX3CR1, IBA-1, TMEM119, and Trem2) and other immune cells (B220, CCR2, CD11b, CD11c, CD138, CD19, CD4, CD45, CD8a, CX3CR1, Ly6C, and NK1.1) (Appendix A). Finally, cells were incubated with Fluorogold to distinguish between live and dead cells. Sample acquisition was performed on a BD LSR Fortessa X-20 Cell Analyzer (ARAFlowCore, Alfred Research Alliance, Monash University, Melbourne, Australia), and collected data were analyzed using FlowJo software (v. 10.6, BD Biosciences, Franklin Lakes, NJ, USA). The numbers of microglia and microglial forward and side scatter were expressed relative to control animals. The expression of microglial markers was determined by geometric mean fluorescence intensity (gMFI), which was normalized by dividing the value of each sample by the mean value of the C57BL/6 microglia in each experiment, which enabled the pooling of data from multiple experiments.

### 2.8. Intracellular Cytokine Flow Cytometry

The number of cytokine-positive microglia following ex vivo lipopolysaccharide (LPS) stimulation was determined through intracellular cytokine flow cytometry. Brains were collected, and a single-cell suspension was prepared using the same method for flow cytometry. Cells were resuspended in RPMI (Thermo Fisher Scientific), 1% Penicillin-Streptomycin (Thermo Fisher Scientific), and 1:1000 Golgi plug (BD Biosciences) and placed into 2 individual wells of a 24-well plate at a concentration of 1 × 10^6^ cells per well. One well from each sample was stimulated with 10 ng/mL LPS (Sigma Aldrich) for 4 h at 37 °C in 5% CO_2_. Stimulation was stopped with cold RPMI media, and cells were collected and then stained for 20 min at 4 °C with receptor-specific antibodies to identify microglial cells (CD45^+^CD11b^+^TMEM119^+^). Cells were then fixed with 2% PFA for 30 min at 4 °C and permeabilized with 1× BD Perm/Wash (BD Biosciences) for 15 min at room temperature. Fixed and permeabilized cells were incubated in 1× BD Perm/Wash buffer containing anti-TNF-α (Thermo Fisher Scientific) or IgG isotype control (Appendix A) at 4 °C for 30 min. Samples were resuspended and acquired immediately on a BD LSR Fortessa X-20 Cell Analyzer, and data were analyzed using FlowJo software (v. 10.6, BD Biosciences). Microglia were designated as CD45^+^CD11b^+^TMEM119^+^. Cytokine positive microglia were identified after gating out stimulated isotype control events, and data were expressed as a ratio of total microglia relative to the average unstimulated C57BL/6 microglia from each experiment.

### 2.9. Quantitative Real-Time Polymerase Chain Reaction (qPCR)

Fresh brains were collected for qPCR to evaluate inflammatory gene expression. The cortex was processed for RNA isolation using the RNeasy mini kit (Qiagen, Hilden, Germany) and a QIAcube (Qiagen), as per the manufacturer’s protocol. Eluted RNA concentration and purity were determined using a Qiagen QIAexpert spectrophotometer. One microgram of eluted RNA from each sample was reversed transcribed into cDNA using a QuantiTect Reverse Transcription kit (Qiagen) and diluted to 1:10.

Samples were pipetted into a 192-well plate using a Qiagility liquid handling robot (Qiagen). Samples were run in duplicate, and cDNA was amplified with a QuantStudio 7 Flex Real-Time PCR system (Applied Biosystems, Waltham, MA, USA). TaqMan fast advanced gene expression assays (Thermo Fisher Scientific) were used for quantification of gene expression, as outlined in Appendix A.

Relative gene expression ratios were calculated using the 2^−ΔΔCT^ method and normalized to the geometric mean of the housekeeping genes (*Ppia* and *Hprt*), which we have previously shown to be stable in the brains of C57BL/6 mice [47]. The expression of each gene was normalized to control mice.

### 2.10. Statistical Analysis

Statistical analysis was performed using GraphPad Prism (ver. 8.2.1, GraphPad Software, Boston, MA, USA) with statistical significance reported as *p* < 0.05. All quantitative data passed normality tests (Anderson–Darling, D’Agostino and Pearson, Shapiro–Wilk and Kolmogorov–Smirnov); thus, unpaired T-tests were used to examine the factor of genotype in SHIP-1-deficient or Lyn-deficient mice compared to their respective controls. Where appropriate, a two-way ANOVA test was used to examine the factors of genotype and time in behavior tests, and factors of genotype and LPS treatment in stimulation studies. A two-way ANOVA test was also conducted to examine the factors of genotype and sex in behavior, body weight, and spleen weight. However, examination of potential differences between sexes in other measurements were not statistically analyzed due to the lower numbers of animals of each sex and the unavoidable uneven distribution of sexes between experimental groups. Data are presented as mean ± standard error of the mean (SEM). Females are presented as solid data points, whereas males are presented as clear data points.

## 3. Results

### 3.1. Subsection

#### 3.1.1. Young Adult Lyn−/− Mice Have Normal Morphological Characteristics and Behavioral Phenotype

To examine the role of Lyn in the steady-state brain, young adult Lyn-deficient mice were assessed. At necropsy, body weight and spleen weight were recorded, while brain volumes were measured to grossly examine whether Lyn deficiency influenced neuroanatomy (Figure 1). Lyn−/− mice had comparable body weights to control mice (Figure 1a; t_32_ = 0.48, *p* > 0.05), and spleen weights, expressed relative to body weight, were also similar (Figure 1b; t_32_ = 0.99, *p* > 0.05). While it is well known that aged Lyn-deficient mice develop splenomegaly due to an expansion of myeloid cells as the mice develop lupus manifestations, young adult mice have a normal spleen weight albeit with an altered immune cell composition [48]. Post hoc analysis revealed that body weight was modestly reduced in young adult female Lyn−/− mice compared to male Lyn−/− mice, which almost reached statistical significance (post hoc analysis, *p* = 0.0505). Volumetric analyses of brain slices (Figure 1c) showed that the total volume of the sectioned brain, the volume of the dorsal cortex, and the hippocampal volume were not different between Lyn−/− and control mice (Figure 1d; t_11_ = 0.63, t_11_ = 0.85, t_11_ = 1.62, *p* > 0.05).

Behavioral phenotypes of Lyn−/− mice were also assessed against age-matched control mice. In the open-field test, all animals spent a comparable proportion of time in the center zone of the arena (Figure 1e; t_35_ = 1.76, *p* = 0.087). Similarly, the number of entrances into the center zone, the total distance traveled, and the average velocity were unchanged in Lyn−/− mice, indicating normal locomotor and exploratory activity in this strain (Figure 1f–h; t_35_ = 1.17, t_35_ = 0.97, t_35_ = 0.31, *p* > 0.05). Likewise, the time spent in the open arms in the Elevated Plus Maze and the distance traveled in this task were not altered by Lyn deficiency (Figure 1i,j; t_35_ = 0.43, t_35_ = 0.68, *p* > 0.05). On the accelerating rotarod, the latency to fall during the trial phase and test phase was also comparable between genotypes, while both groups showed learning over time (increased time on the rotarod between trials) (Figure 1k; F_1,70_ = 10.65, *p* = 0.002, main effect of time; *p* > 0.05, effect of genotype). Lastly, there were no differences in the ability to discern the novel arm from the familiar arm during the Y-maze test phase, as demonstrated by the comparable discrimination index between genotypes (Figure 1l; t_35_ = 0.82, *p* > 0.05).

#### 3.1.2. Microglia Activation Was Unchanged in Lyn-Deficient Mice

Given that Lyn expression is reportedly enriched in mouse microglia [21,25], where it is thought to play an inhibitory role [31], we next examined whether Lyn deficiency influenced steady-state microglial activation by measuring IBA-1 expression, microglia numbers, and microglial morphology (Figure 2). In the cortex and dentate gyrus of one hemisphere, IBA-1 expression was comparable between genotypes (Figure 2a–c; t_11_ = 0.73, t_11_ = 1.34, *p* > 0.05). Furthermore, the number of IBA-1+ microglia within both regions was unchanged in Lyn−/− mice (Figure 2d,e; t_11_ = 1.02, t_11_ = 0.30, *p* > 0.05). In addition, the number of branches, average branch length, and soma size were comparable between control and Lyn−/− mice (Figure 2f–h; t_11_ = 0.67, t_11_ = 0.97, t_11_ = 1.39, *p* > 0.05). Collectively, these analyses indicate that Lyn deficiency does not affect the homeostatic microglial activation state.

#### 3.1.3. Lyn Deficiency Does Not Alter the Steady-State Microglial Phenotype

To further characterize Lyn-deficient microglia, flow cytometry was conducted on dissociated whole brain tissue to examine microglial cell surface phenotype (Figure 3). Microglia were identified by selecting live CD45^+^ cells and then gating on CD11b^+^ TMEM119^+^ cells (Figure 3a). The total number of microglia and their size and complexity, as represented by forward and side scatter, respectively, were comparable between genotypes (Figure 3b–d; t_18_ = 1.08, t_18_ = 0.28, t_18_ = 0.48, *p* > 0.05). In addition, the expressions of activation markers on microglia such as IBA-1, CX3CR1, and CD45 were unaffected by Lyn deficiency (Figure 3e–g; t_18_ = 0.84, t_18_ = 0.16, t_18_ = 1.00, *p* > 0.05). Furthermore, expression of the myeloid receptor that senses tissue damage, TREM2, was also unaffected in Lyn−/− mice (Figure 3h; t_18_ = 0.005, *p* > 0.05). Interestingly, Lyn-deficient microglia had increased expression of the co-stimulatory marker CD86 (Figure 3i; t_18_ = 3.86 *p* = 0.001), but they exhibited similar expression of the anti-inflammatory marker CD206 (Figure 3j; t_18_ = 1.98 *p* = 0.063) and homeostatic markers CD14 and TMEM119 to control microglia (Figure 3k,l; t_18_ = 1.02, t_18_ = 0.30, *p* > 0.05). In addition, the numbers of CD4 and CD8 T cells, B cells, and myeloid cells in Lyn−/− brains were similar to control mice (Appendix A). Together, the steady-state microglial phenotype was not significantly affected by the absence of Lyn and there was no evidence of an aberrant influx of peripheral immune cells into the brain.

#### 3.1.4. Lyn Deficiency Does Not Alter Expression of Neuroimmune Response Genes

To examine whether Lyn deficiency altered neuroimmune responses and neuroinflammation, qPCR was conducted on cortical tissue (Figure 4). As expected, Lyn was detectable in the control strain but not in Lyn−/− mice (Figure 4a). Gene expression of the pro-inflammatory cytokine *Il1β* was unchanged in Lyn−/− mice compared to control animals (Figure 4b; t_15_ = 1.35, *p* > 0.05). The expressions of genes encoding the pro-inflammatory marker Fcγr3 (CD16) and microglial markers Iba-1 and Tmem119 were unchanged between genotypes (Figure 4c–e; t_19_ = 0.96, t_18_ = 0.69, t_18_ = 0.46, *p* > 0.05). Furthermore, the expression of *Trem2* was comparable between Lyn−/− and control mice (Figure 4f; t_18_ = 1.25, *p* > 0.05). Lastly, there were no differences in the expression of genes encoding the astrocyte marker Gfap or chemokine Ccl2 in Lyn−/− brain tissue (Figure 4g,h; t_18_ = 0.32, t_18_ = 0.15, *p* > 0.05). Therefore, deficiency of Lyn had minimal effect at the steady state on the expression of genes associated with neuroinflammatory and neuroimmune responses.

Given that Lyn is not the only Src kinase expressed in microglia, the expression of other family members was analyzed to assess if there was compensation for the lack of Lyn expression. However, gene expression of *Fyn*, *Src*, and *Hck* was unchanged in Lyn−/− mice (Figure 4i–k; t_20_ = 1.75, t_20_ = 1.56, t_17_ = 0.87, *p* > 0.05).

Together, the above-described results indicate that deficiency of the immune cell negative regulator Lyn had no significant impact on microglial phenotype or neuroinflammation at the steady state. Thus, the focus of the study was switched to SHIP-1, which is another negative regulator of immune cell signaling. While Lyn and SHIP-1 have overlapping functions in controlling immune cell signaling [10,49,50,51], the more exaggerated myeloid cell phenotype of SHIP-1-deficient mice [12,13,14,50] led us to hypothesize that it may play a broader role in controlling microglial activation.

#### 3.1.5. Brain Volume of SHIP-1−/− Mice Is Unchanged but SHIP1−/− Mice Display Behavioral Differences

To confirm the SHIP-1-deficient phenotype, body weight and spleen weight were first assessed (Figure 5). As expected, SHIP-1−/− mice had reduced body weights (Figure 5a; t_65_ = 3.40, *p* = 0.001) and increased spleen-to-body weight ratios compared to age-matched control mice [12,13,14] (Figure 5b; t_57_ = 11.90, *p* < 0.0001). The early-onset splenomegaly phenotype of SHIP-1-deficient mice is driven by a marked expansion of myeloid cells and nucleated erythroid cells as well as plasma cells, together with a loss of mature B cells [12]. Nonetheless, volumetric analysis of the brains of SHIP-1−/− mice showed no difference to controls in the size of the whole sectioned brain, dorsal cortex, or dorsal hippocampus (Figure 5c,d; t_14_ = 0.25, t_14_ = 0.50, t_14_ = 0.50, *p* > 0.05).

Next, SHIP-1−/− mice and littermate controls underwent a battery of behavior analyses to test for differences in behavioral phenotypes. In the open-field test (Figure 5e), both genotypes spent the same amount of time in the center zone of the arena, indicating comparable anxiety-like behavior (Figure 5f; t_37_ = 0.92, *p* > 0.05). However, compared to controls, SHIP-1−/− mice showed a reduced number of entries into the center zone (Figure 5g; t_37_ = 2.47, *p* = 0.018), a reduced total distance traveled (Figure 5h, t_37_ = 3.35, *p* = 0.002), and a reduced average velocity (Figure 5i; t_37_ = 3.47, *p* = 0.001), indicating reduced exploratory and locomotor activity. In the Elevated Plus Maze, both groups spent comparable time in the open arms and traveled similar distances during testing (Figure 5j,k; t_37_ = 0.34, t_37_ = 1.07, *p* > 0.05). Both groups also performed similarly on the accelerating rotarod, with expected improvement over time, but no differences between genotypes were observed (Figure 5l; main effect of time F_1,26_ = 23.43 *p* < 0.0001). Lastly, there was no difference in the discrimination index in the Y-maze between genotypes, indicating comparable working memory performance (Figure 5m; t_37_ = 0.64, *p* > 0.05).

#### 3.1.6. SHIP-1 Deficiency Altered Microglial Phenotype in the Cortex and Hippocampus

As SHIP-1 expression is enriched in microglia [28,29], the microglial activation status in the cortex and hippocampal dentate gyrus of SHIP-1−/− mice was investigated by determining the IBA-1 fluorescence intensity, the number of microglia, and the microglial morphology (Figure 6). The fluorescence intensity of IBA-1 staining (Figure 6a) was elevated in the cortex and dentate gyrus of SHIP-1−/− mice compared to controls (Figure 6b; t_14_ = 3.63, *p* = 0.003; Figure 6c; t_14_ = 2.65, *p* = 0.019). Interestingly, the number of IBA-1+ microglia in the cortex was unchanged between genotypes (Figure 6d; t_14_ = 1.83, *p* = 0.088), whereas the number was increased in the dentate gyrus of SHIP-1−/− mice (Figure 6e; t_14_ = 2.31, *p* = 0.037). Assessment of microglial morphology showed that the average branch length, number of branches, and cell body size of cortical microglia were unchanged between SHIP-1−/− mice and controls (Figure 6f–h; t_14_ = 1.65, t_14_ = 0.92, t_14_ = 1.74, *p* > 0.05). Collectively, these results indicate that SHIP-1-deficient microglia exhibited a modestly activated phenotype compared to those of control brains, although this did not manifest in a noticeable change in morphology.

#### 3.1.7. SHIP-1-Deficient Microglia Exhibit an Altered Cell Surface Phenotype

Since SHIP-1-deficient microglia exhibited a modestly activated phenotype by immunofluorescence, flow cytometry was next performed to characterize microglial phenotypes of the whole brain of SHIP-1-deficient mice (Figure 7) using the gating strategy previously outlined in Figure 3a. The number of microglia and their forward scatter profile were comparable between genotypes (Figure 7a,b; t_24_ = 0.69, t_24_ = 1.24, *p* > 0.05); however, microglia from SHIP-1−/− mice had reduced side scatter (Figure 7c; t_24_ = 2.71, *p* = 0.012). In addition, microglia from SHIP-1−/− mice had elevated expression of the activation marker CD45 (Figure 7d; t_24_ = 3.53, *p* = 0.002) but unchanged CX3CR1 expression (Figure 7e; t_24_ = 0.97, *p* > 0.05). Similarly, microglia from SHIP-1−/− mice had elevated expression of the costimulatory marker CD40 (Figure 7f; t_24_ = 2.97, *p* = 0.007), but comparable expression of TREM2 to control mice (Figure 7g; t_24_ = 0.78, *p* > 0.05). There was a trending increase in microglial expression of CD86 (Figure 7h; t_24_ = 1.90 *p* = 0.069), while expression of CD206, TMEM119 and CD14 were comparable between genotypes (Figure 7i–k; t_24_ = 0.94, t_24_ = 1.68, t_24_ = 1.05, *p* > 0.05). The number of CD4 and CD8 T cells, B cells, neutrophils, and macrophages in the brain was also determined, and no differences were observed between genotypes (Appendix A). Taken together, these results indicate that microglia from the steady-state brain of SHIP-1−/− mice displayed a unique phenotype comprised of increased expression of several cell surface markers associated with activation but there was no evidence of peripheral leukocytes trafficking into the brain.

#### 3.1.8. Increased Inflammatory Gene Expression in Cortical Brain Tissue of SHIP-1-Deficient Mice

Given the changes observed in microglial morphology and cell surface marker expression in SHIP-1−/− mice, gene expression in cortical brain tissue was next assessed to determine if these reflected changes in glial and immune responses (Figure 8). As expected, while *Inpp5d* was detectable in the cortex of control mice, SHIP-1−/− mice had no *Inpp5d* expression, confirming their genotype (Figure 8a). Expression of the gene encoding the microglial activation marker Iba1 was significantly increased in SHIP-1−/− mice (Figure 8b; t_10_ = 3.16, *p* = 0.031) in line with the previous immunofluorescence analyses (Figure 6a-c), whereas expression of the homeostatic marker *Tmem119* was comparable between genotypes (Figure 8c; t_10_ = 0.23, *p* > 0.05). Interestingly, cortical brain tissue from SHIP-1−/− mice had increased gene expression of the pro-inflammatory marker *Fcγr3* (*Cd16*) compared to controls (Figure 8d; t_10_ = 2.51, *p* = 0.031), while *Cd86* and *Cd206* were unchanged (Figure 8e,f; t_9_ = 1.05, t_10_ = 1.83, *p* > 0.05). SHIP-1−/− mice also exhibited increased expression of the astrogliosis marker *Gfap* (Figure 8g; t_10_ = 2.58, *p*= 0.027) and monocyte chemokine *Ccl2* (Figure 8h; t_9_ = 4.22, *p* = 0.002). Collectively, these studies show that the steady-state expression of genes related to neuroinflammatory responses is elevated in the cortex of SHIP-1−/− mice.

#### 3.1.9. Microglia from Lyn−/− and SHIP-1−/− Mice Are Hyper-Responsive to LPS Stimulation

Thus far, we report that the steady-state brain of SHIP-1−/− mice shows evidence of microglial activation, which was not readily apparent in Lyn−/− mice. Given that both Lyn−/− and SHIP-1−/− mice and their macrophages are reported to be hyper-responsive to LPS challenge [52,53,54], we next examined whether ex vivo microglia from Lyn−/− or SHIP-1-deficient mice would similarly show an exacerbated response to LPS. Single-cell suspensions of brain from Lyn−/− and SHIP-1−/− mice and their respective controls were stimulated with LPS, and flow cytometry was conducted to measure the number of microglia producing the prototypical pro-inflammatory cytokine TNF-α (Figure 9a–c). LPS stimulation significantly increased the number of TNF-α-producing microglia in both Lyn−/− and control mice (Figure 9b,d; effect of genotype, F_1,36_ = 2.88, *p* < 0.0001; effect of stimulation, F_1,36_ = 20.34, *p* < 0.001). A subsequent post hoc analysis revealed an increased number of TNF-α-producing Lyn−/− microglia compared to the control (*p* = 0.003). Similarly, LPS stimulation increased the number of TNF-α-producing microglia in SHIP-1−/− and control mice (Figure 9c,e; effect of genotype, F_1,30_ = 19.17, *p* = 0.0020; effect of stimulation, F_1,30_ = 6.06, *p* = 0.0198), and a post hoc analysis indicated the number of TNF-α-producing microglia was significantly higher in SHIP-1−/− mice (*p* = 0.003). Together, these findings demonstrate that microglia from both Lyn-deficient and SHIP-1-deficient mice are capable of exhibiting enhanced cytokine responses upon LPS stimulation, despite differences in their roles in microglial homeostasis in the steady-state brain.

## 4. Discussion

Previous studies have identified Lyn and SHIP-1 as critical negative regulators of signaling in bone-marrow-derived immune cells, whereby their disruption can drive immunological diseases such as lupus and lung inflammation [11,12,13,14]. As both Lyn and SHIP-1 are expressed in microglia, we sought to define their roles in brain homeostasis by examining Lyn−/− and SHIP-1−/− mice. Together, our findings suggest that SHIP-1 contributes to regulation in the steady-state brain, while Lyn appears dispensable.

### 4.1. Lyn Deficiency Did Not Impact Neuroimmune Responses

Lyn expression is distributed throughout the mammalian brain and has been linked to neural plasticity [20,21,22,23,24]. However, brain volume was unchanged in Lyn-deficient mice, suggesting that brain growth was not affected by the absence of Lyn. Similarly, Lyn-deficient mice did not exhibit significant changes in a comprehensive battery of behavioral assays for anxiety-like behaviors, explorative tendency, motor functioning, or working memory when compared to control animals. Lyn has previously been reported to negatively regulate glutamate receptor signaling and dopamine release in rodents [23,24,55], and one study found that global ablation of *Lyn* in mice was associated with aberrant glutamate signaling and a transient reduction in spontaneous motor activity [21]. The disparity in behavioral outcomes could be attributed to strain variation, differences in the behavioral tests employed between studies, and measurement duration. Furthermore, it should be noted that differences in motor activity between animals in the previous study were dissipated by day 3 of testing, suggesting that anxiety or fear in response to the new testing environment may have been attributed to their behavioral phenotype [56,57].

Previous in vivo and ex vivo studies have shown that Lyn regulates microglial activation in animal models of Alzheimer’s disease [32,58]. Here, however, we failed to identify any changes in microglial phenotype at the steady state in the absence of Lyn when examining microglial morphology and number and the expression of defined microglial activation markers. Phenotypically, microglia in Lyn-deficient mice were subtly skewed towards a more pro-inflammatory state, based on a modest increase in CD86 surface expression, which has previously been identified in other rodent disease models [59,60,61]. This could be driven by an increased sensitivity to surrounding cytokines, akin to what has been reported in Lyn-deficient macrophages [10,52,62]. Importantly, an increase in sensitivity of Lyn-deficient microglia was demonstrated after LPS stimulation, which showed their propensity to over-produce the pro-inflammatory cytokine TNF-α. Together, these findings suggest that Lyn expression is not central to microglial activity during steady state, but may modulate microglial responses to environmental stimuli, such as in the presence of neurotoxic plaques in the context of Alzheimer’s disease as previously described [33,34,35].

As Lyn is critical for regulating the immune system outside the brain [15,62,63], broader neuroimmune responses were assessed in young adult Lyn-deficient mice. These analyses demonstrated similar inflammatory gene expression between Lyn-deficient and control mice, despite the presence of Lyn being reported to regulate inflammatory responses in macrophages such as suppressing inflammasome activities [64]. Similarly, the gene expression of other key Src tyrosine kinases potentially involved in neuroimmune responses [65,66] remained unaltered in Lyn−/− mice, but additional examination of their tyrosine kinase activity may uncover compensatory mechanisms in place in the absence of Lyn regulation.

### 4.2. SHIP-1 Deficiency Exacerbated Neuroinflammation and Microglial Activation

Polymorphisms in the *INPP5D* gene encoding SHIP-1 have been linked to the development of neurodegenerative diseases [36,37,38,39,40]. Further, as SHIP-1 expression is largely confined to microglia in the brain, it appears that SHIP-1 regulation of microglial activity is critical during disease pathogenesis [3]. In the steady-state brain, we found that brain volumes in SHIP-1-deficient mice were not altered compared to controls, suggesting that SHIP-1 expression in the brain is not vital for brain growth. SHIP-1−/− mice displayed a behavioral phenotype characterized by reduced exploratory activity in the open-field test. This was in the context of normal motor function on the rotarod and comparable anxiety-like behavior and working memory function to control mice, indicating that their reduced activity is not due to motor impairments or cognitive or psychiatric deficits. Rather, we propose that the reduced locomotor activity seen in SHIP-1−/− mice is likely attributed to their peripheral inflammatory phenotypes such as ileitis and lung inflammation [16], rather than specific neurological deficits.

Analysis of glial populations in SHIP-1-deficient mice revealed that microglia had higher cell surface expression of markers such as IBA-1, CD45, and CD40 [67,68]. In addition, gene expression of *Fc*γ*r3* was increased in SHIP-1-deficient mice, which has previously been associated with a shift in microglial phenotype following spinal cord injury [69,70]. These markers are all indicative of an activated phenotype. Nonetheless, morphological analyses revealed SHIP-1−/− microglia did not display a shift in morphology, but instead exhibited a surveillant or inactive form similar to microglia of control mice [71].

Interestingly, flow cytometry revealed that microglia from SHIP-1-deficient mice had reduced side scatter, likely denoting reduced granularity, and potentially implicating a change in phagocytosis. An increase in side scatter has been observed in phagocytosis studies [72]; however, this likely reflects the type of assay used, which involves phagocytosis of fluorescent particles. A previous study observed increased phagocytosis by BV2 microglia when both SHIP-1 and SHIP-2 activity were pharmacologically inhibited [28]. Additionally, recent findings suggested that conditional deletion of *Inpp5d* in microglia in Alzheimer’s disease models augmented phagocytosis, increasing plaque encapsulation and engulfment by microglia [73,74]. We have shown that gene expression of the low-affinity phagocytic receptor involved in clearing immune complexes, FcγR3/CD16, is increased in the brain tissue of SHIP-1-deficient mice. SHIP-1 is implicated in regulating signaling from FcγR3 in natural killer cells, where it has been shown to limit FcγR3-dependent cytotoxic function [75]. It would now be interesting to assess the activity of FcγR3 in microglia of SHIP-1-deficient mice. SHIP-1-deficient mice also develop antibody-mediated autoimmune disease [12] and likely have circulating immune complexes, although there is no evidence that these are present in the brain of young adult mice. However, we observed increased CD40 expression in SHIP-1-deficient microglia, suggesting heightened antigen presentation function which could alternatively suppress phagocytosis [76,77]. Furthermore, increased CD40 expression has been linked with several neurodegenerative diseases and observed in microglial populations around Aβ plaques [78,79]. Therefore, further examination of the regulatory role of SHIP-1 in microglial phagocytosis is required.

Previous studies have shown that SHIP-1 deficiency affects the hematopoietic stem cell compartment, leading to a substantial increase in the number of hematopoietic progenitors [12,13,14,50]. This drives splenomegaly and leads to the infiltration of myeloid cells into the lung parenchyma and lamina propria of the ileum [13,18,19,50]. However, this was not the case in the steady-state brain, at least at a young adult age, as both histological and flow cytometry analyses revealed similar numbers of microglia in SHIP-1-deficient and control mice and no abnormal leukocytic infiltration. Given that microglial populations are largely maintained independently of hematopoietic stem cells, and the blood–brain barrier acts to prevent leukocyte infiltration into the brain under homeostatic conditions [80,81], these studies suggest that homeostatic microglial proliferation was largely unaffected by the absence of SHIP-1 and it appears that it is sustained independently of peripheral circulating macrophages.

SHIP-1 is also reported to negatively regulate macrophage activation against inflammatory stimuli [53,82]. Herein, TNF-α production was exacerbated in SHIP-1-deficient microglia in response to an LPS challenge, likely as a result of an increased sensitivity to LPS as the Toll-Like Receptor 4 transduces signals through the PI3K-AKT pathway [3,83]. Furthermore, in combination with microglial populations exhibiting an increased activated phenotype at baseline, SHIP-1 deficiency may promote a cumulative effect in microglial responses, similar to the “two-hit” hypothesis, see [84]. To the best of our knowledge, this is the first ex vivo examination of the cytokine responsiveness of SHIP-1-deficient microglia, as current studies are limited to immortal cell lines (BV2 microglia), and there are conflicting reports on whether SHIP-1 deficiency suppresses or promotes microglial cytokine production [28,85]. Collectively, our findings suggest that SHIP-1 restrains microglial activation, and given that chronically activated microglia and augmented brain cytokines are considered pathogenic for neurodegenerative diseases and psychiatric disorders [86], further assessment into how SHIP-1 regulates these processes is warranted.

### 4.3. Limitations and Future Directions

The elevated neuroinflammation in SHIP-1-deficient mice may be in part perpetuated by their augmented peripheral inflammatory responses [12,13,87]. Although serum cytokines were not measured in this study, we and others have previously shown that young adult SHIP-1−/− mice on a C57BL/6 background exhibit increased circulating cytokines, such as IFNγ, TNFα, IL-4, IL-6, G-CSF, and GM-CSF [12,88]. This may alter the integrity of the blood–brain barrier and lead to increased cytokine levels in the brain [89,90], ultimately enhancing neuroimmune responses and obscuring the effect of the intrinsic loss of SHIP-1 activity in microglia. Therefore, future studies utilizing a combination of conditional knockout models and ex vivo cell culture experiments are needed to fully define the intrinsic role of SHIP-1 in microglia.

Furthermore, as the microglia of Lyn-deficient mice at the steady state showed minimal changes, this suggests that other Src family kinases or alternative Lyn-regulated pathways like p38-MAPK might be operative, which should now be investigated. Nonetheless, cytokine production by LPS-stimulated microglia was increased by Lyn deficiency, suggesting that factors that alter *Lyn* gene expression or Lyn activity may be pathogenic in neurodegenerative diseases driven by microglia.

Another limitation of this study was the unequal distribution of males and females in each group, prohibiting the analysis of both sexes separately and consideration of sex as a biological variable during statistical analysis. From unpublished observations, female SHIP-1-deficient mice may be more susceptible to peripheral inflammatory disease than male mice, which may exacerbate their neuroinflammatory response, but further investigation is required to elucidate the effects of sex on glial responses and neuroinflammation in the SHIP-1-deficient brain.

Finally, our study was limited to a single time point of 11-12 weeks of age. Given that peripheral inflammation worsens in an age-dependent manner in SHIP-1-deficient mice [12,13,14], additional analyses of younger mice such as 4 weeks of age, may reveal intrinsic changes in the brain with a reduced influence from peripheral cytokines. This would also diminish the effect of age, as normal aging is associated with increased microglial activation, reduced phagocytosis, and a pro-inflammatory skewed phenotype [91,92]. Conversely, examining older mice may also reveal whether loss of Lyn or SHIP-1 expression impacts microglial responses and inflammatory changes during aging.

## 5. Conclusions

Our findings indicate that SHIP-1, but not Lyn, is involved in restraining microglial activation and neuroinflammation during homeostasis in the young adult brain. Nonetheless, when presented with an immune challenge, both Lyn and SHIP-1 participate in controlling microglial responses, further supporting their increasingly recognized role in modulating responsiveness to immunological stressors. In conclusion, this study suggests that Lyn and SHIP-1 have independent roles in homeostatic microglia but they both function to restrict microglial activation during neuroimmune responses. Collectively, future studies are warranted to further delineate the role of signaling pathways in microglial responses in health and disease, which may lead to their manipulation for potential therapeutic benefits for neurodegenerative diseases.

## Figures and Tables

**Figure 1 cells-12-02378-f001:**
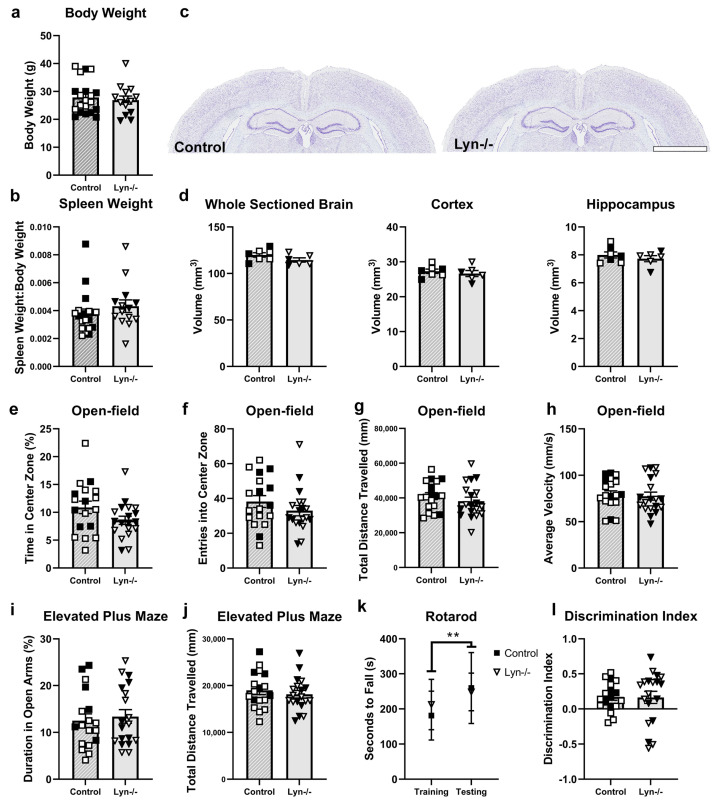
Body weight, spleen weight, brain volume, and behavior were unchanged in Lyn−/− mice. (**a**) Body weight and (**b**) spleen weight of young adult Lyn−/− and control mice (n = 18–19/group). (**c**) Representative images of cresyl-violet-stained brain sections (scale bar = 2 mm), and (**d**) volumetric analysis of total sectioned brain, the dorsal cortex, and the dorsal hippocampus of Lyn−/− and control mice (n = 6–7/group). (**e**) Time spent in center zone; (**f**) entries into center zone; (**g**) total distance traveled, and (**h**) average speed during the open-field test. (**i**) Time spent in the open arms and (**j**) distance traveled during Elevated Plus Maze. (**k**) Latency to fall (sec) during the rotarod test. (**l**) Discrimination index in the Y-maze test, calculated by time spent in familiar arm against time spent in the novel arm. n = 18–19/group for behavior tests. Females = solid, males = clear. Unpaired *t*-test and two-way ANOVA; ** *p* < 0.005 (main effect of time).

**Figure 2 cells-12-02378-f002:**
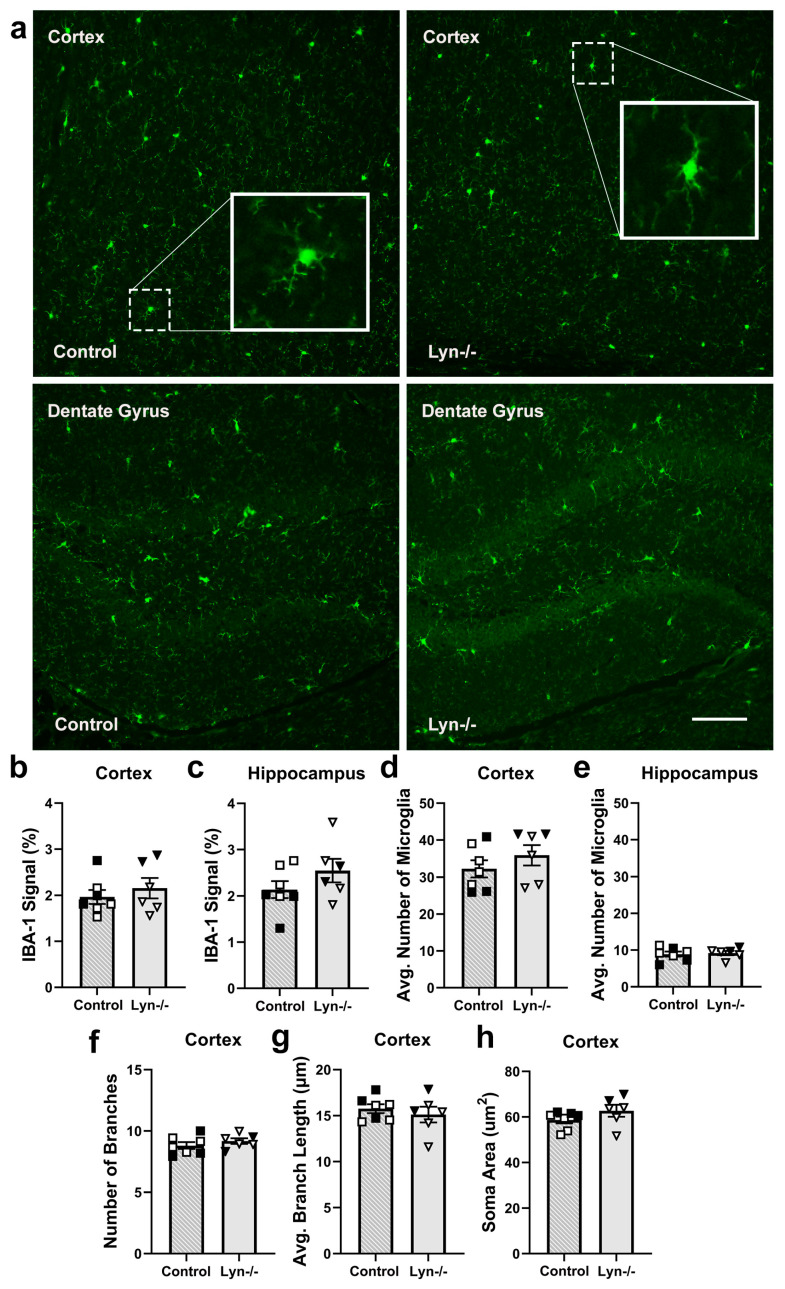
Microglial activation state, number, and morphology were unchanged in Lyn−/− mice. (**a**) Representative immunofluorescence images of IBA-1-stained brains from Lyn−/− and control mice (scale bar = 100 µm). (**b**,**c**) Proportion of IBA-1 staining and (**d**,**e**) number of microglia in the cortex and dentate gyrus. (**f**) The number of branches, (**g**) average branch length, and (**h**) soma area of cortical microglia. n = 6–7/group for histology analysis. Females = solid, males = clear. Unpaired *t*-test.

**Figure 3 cells-12-02378-f003:**
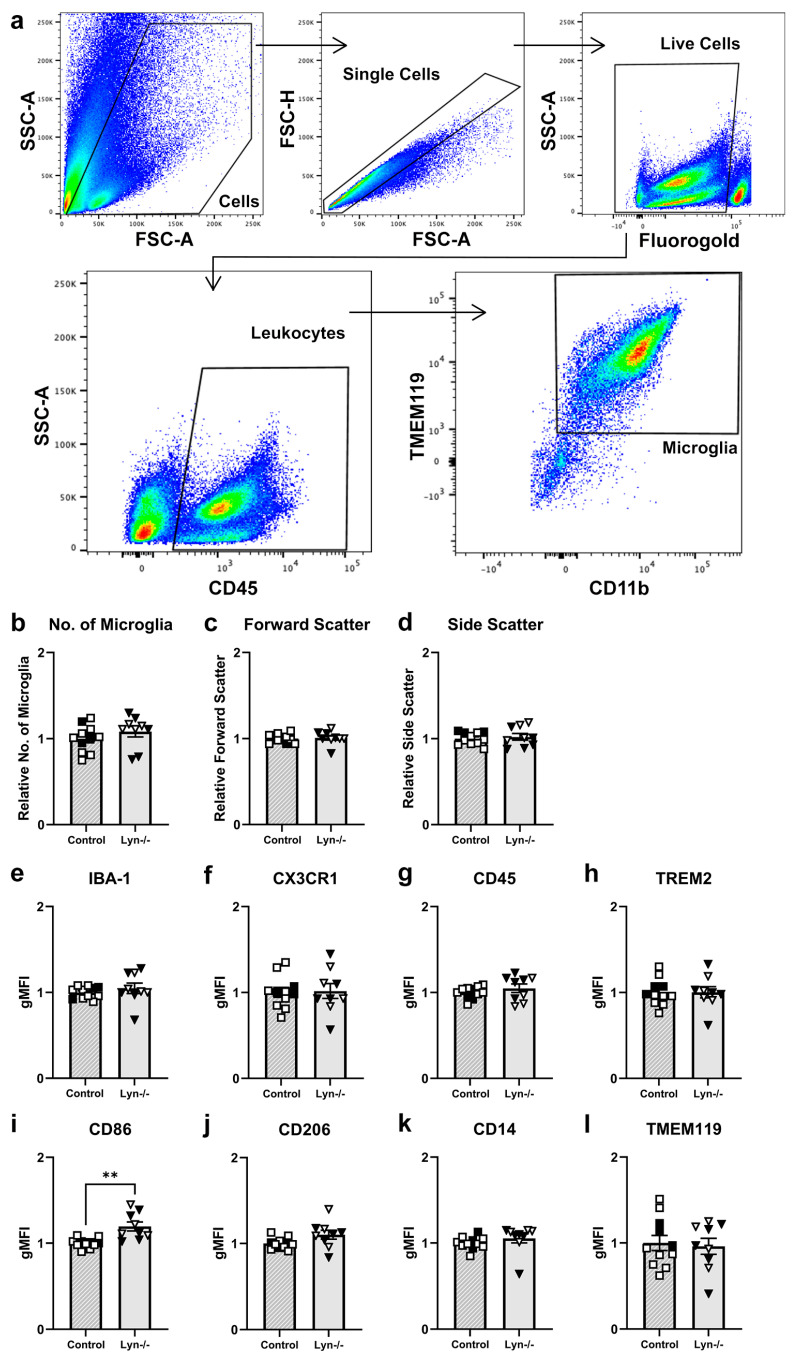
The microglial phenotype was not altered by Lyn deficiency. Single-cell suspensions of brain cells were analyzed by flow cytometry. (**a**) A gating strategy used to distinguish microglia from the whole brain of Lyn−/− and control mice. (**b**) The relative number of microglia and their (**c**) forward scatter and (**d**) side scatter characteristics. (**e**–**l**) Geometric mean fluorescence intensity (gMFI) of cell surface markers on microglia relative to expression on control microglia. n = 9–11/group for flow cytometry. Females = solid, males = open. Unpaired *t*-test. ** *p* < 0.005.

**Figure 4 cells-12-02378-f004:**
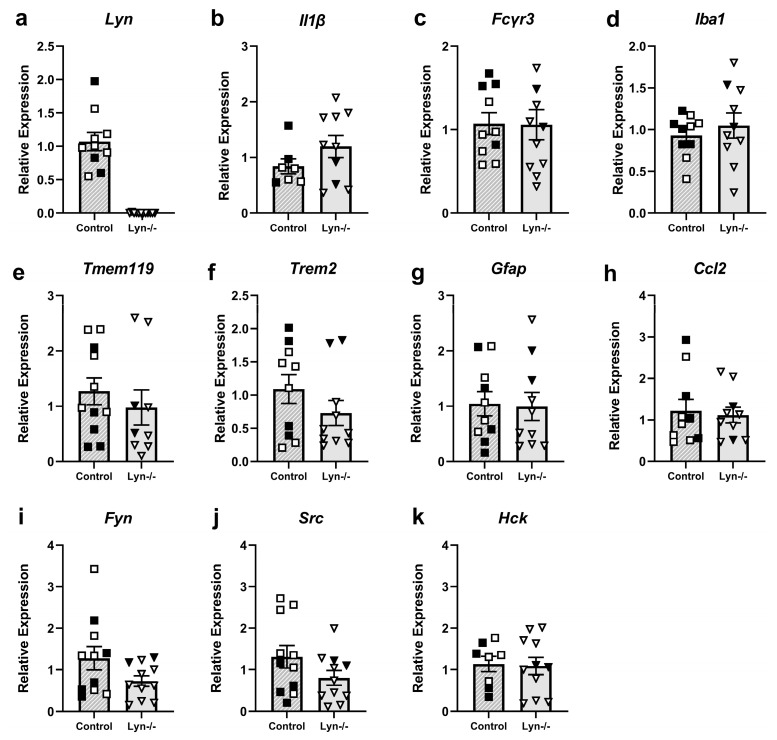
Expressions of immune and inflammatory genes and Src family kinase genes were unchanged in Lyn−/− brain tissue. (**a–k**) Gene expression analysis by qPCR of brain tissue from Lyn−/− and control mice. Gene expression is relative to the average of the control group. n = 7–10/group, expressed relative to control group mean. Females = solid, males = open. Unpaired *t*-test.

**Figure 5 cells-12-02378-f005:**
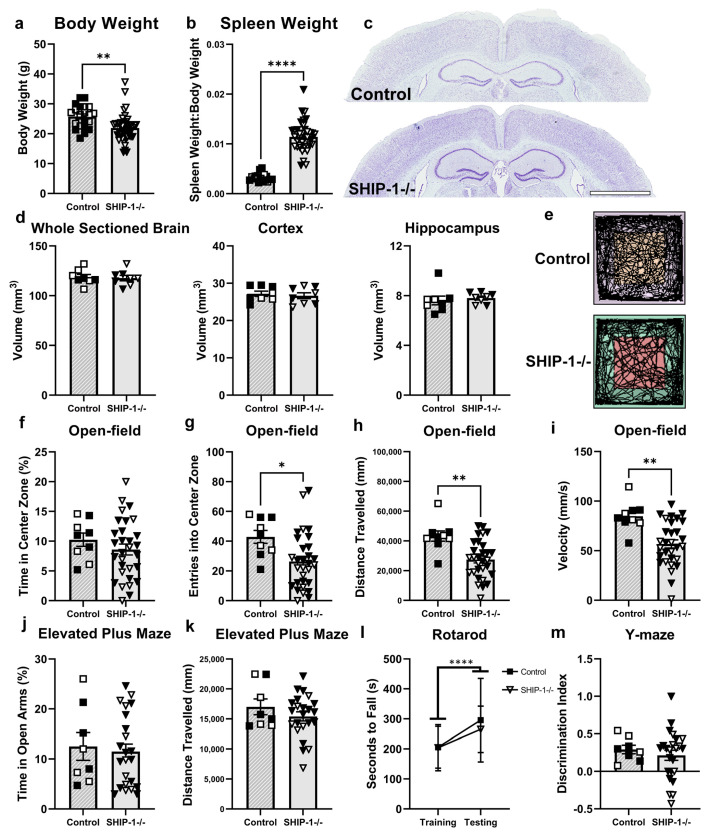
SHIP-1−/− mice had normal brain volume but reduced explorative tendency. (**a**) Body weight and (**b**) spleen weight relative to body weight of young adult SHIP-1−/− mice (n = 21–45/group). (**c**) Representative images of cresyl-violet-stained brain sections (scale bar = 2 mm). (**d**) Volumetric analysis of the sectioned brain, and dorsal cortex and hippocampus of SHIP-1−/− and littermate control mice (n = 8/group). (**e**) Representative Topscan tracing of control and SHIP-1−/− mice during the open-field test. (**f**) Time spent in center zone, (**g**) entries into center zone, (**h**) distance traveled, and (**i**) speed in open-field tests (n = 9 control and 29 SHIP-1−/− mice). (**j**) Time spent in open arms and (**k**) total distance traveled during Elevated Plus Maze (n = 8 control and 23 SHIP-1−/− mice). (**l**) Time (sec) spent on rotarod before falling on consecutive training and testing days (n = 8 control and 23 SHIP-1−/− mice). (**m**) Discrimination index calculated from time spent in novel arm against time spent in familiar arm during Y-maze (n = 8 control and 23 SHIP-1−/− mice). Females = solid, males = open. Unpaired *t*-test and two-way ANOVA. * *p* < 0.05, ** *p* < 0.005, **** *p* < 0.0001.

**Figure 6 cells-12-02378-f006:**
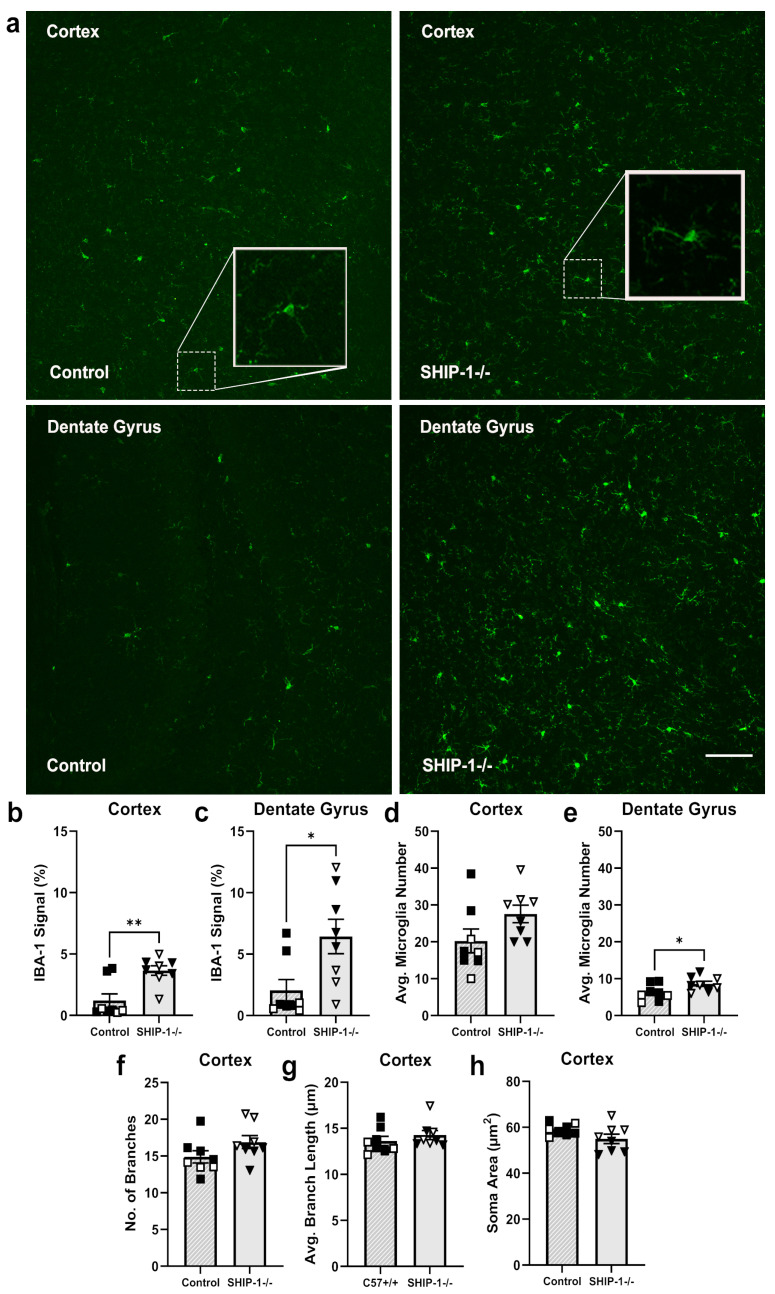
SHIP-1−/− mice exhibited modest changes to microglia. (**a**) Representative images of IBA-1-stained sections of the cortex and dentate gyrus of control and SHIP-1−/− mice (scale bar = 100 µm). Quantitation of IBA-1 fluorescence intensity in the (**b**) cortex and (**c**) dentate gyrus. The average number of IBA-1+ microglia in the (**d**) cortex and (**e**) dentate gyrus. (**f**) The number of branches, (**g**) average branch length, and (**h**) soma area of cortical microglia. n = 8/group for histology analysis. Females = solid, males = open. Unpaired *t*-test. * *p* < 0.05, ** *p* < 0.005.

**Figure 7 cells-12-02378-f007:**
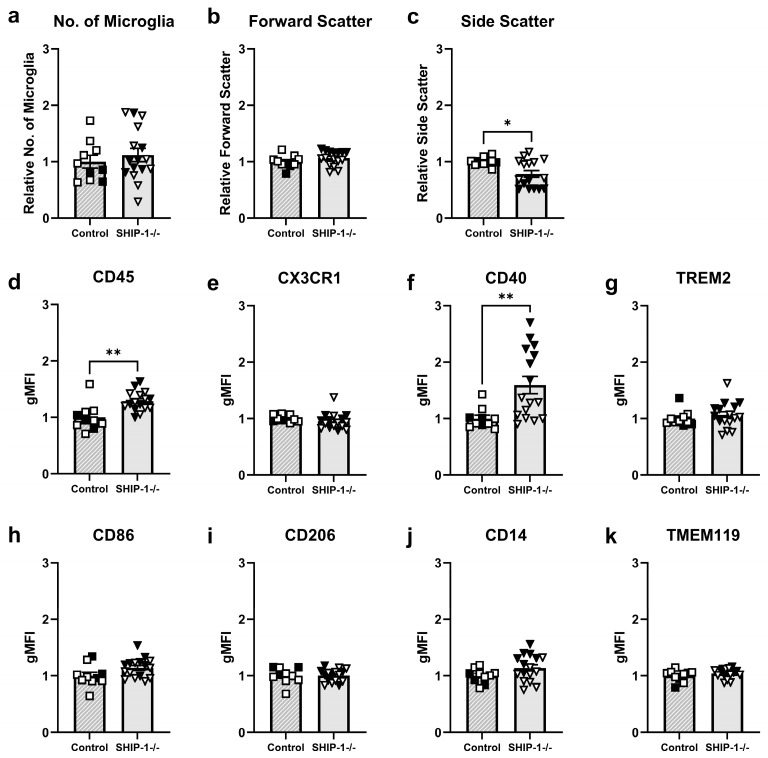
Altered microglial phenotype in SHIP-1−/− mice. Single-cell suspensions of brain cells were analyzed by flow cytometry gating on CD45^+^TMEM119^+^CD11b^+^ microglia. (**a**) The number of microglia and their (**b**) forward scatter and (**c**) side scatter profiles in SHIP-1−/− and control brains. (**d**–**k**) Geometric mean fluorescence intensity (gMFI) of cell surface markers on microglia relative to expression on control microglia. n = 10–16/group for flow cytometry. Females = solid, males = open. Unpaired *t*-test. * *p* < 0.05, ** *p* < 0.005.

**Figure 8 cells-12-02378-f008:**
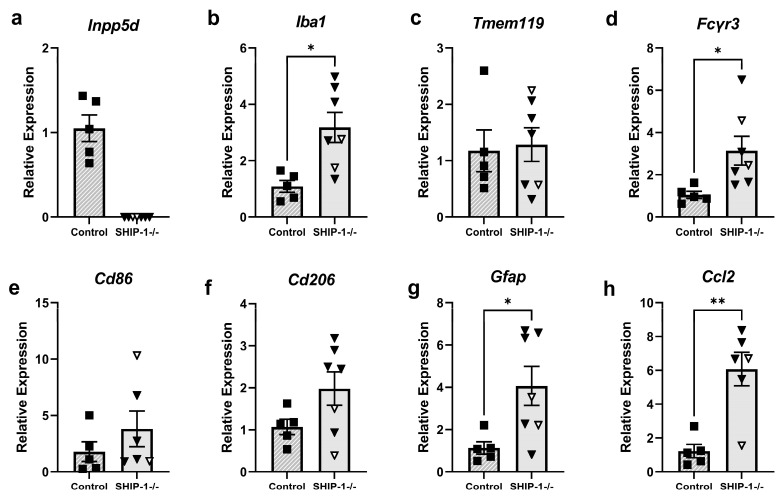
Expression of pro-inflammatory genes was increased in SHIP-1−/− mice. (**a**–**h**) Expression of genes associated with immune and inflammatory responses in cortical brain tissue from control and SHIP-1−/− mice. Gene expression is relative to the average of the control group. n = 5–7/group for gene expression. Females = solid, males = open. Unpaired *t*-test. * *p* < 0.05, ** *p* < 0.005.

**Figure 9 cells-12-02378-f009:**
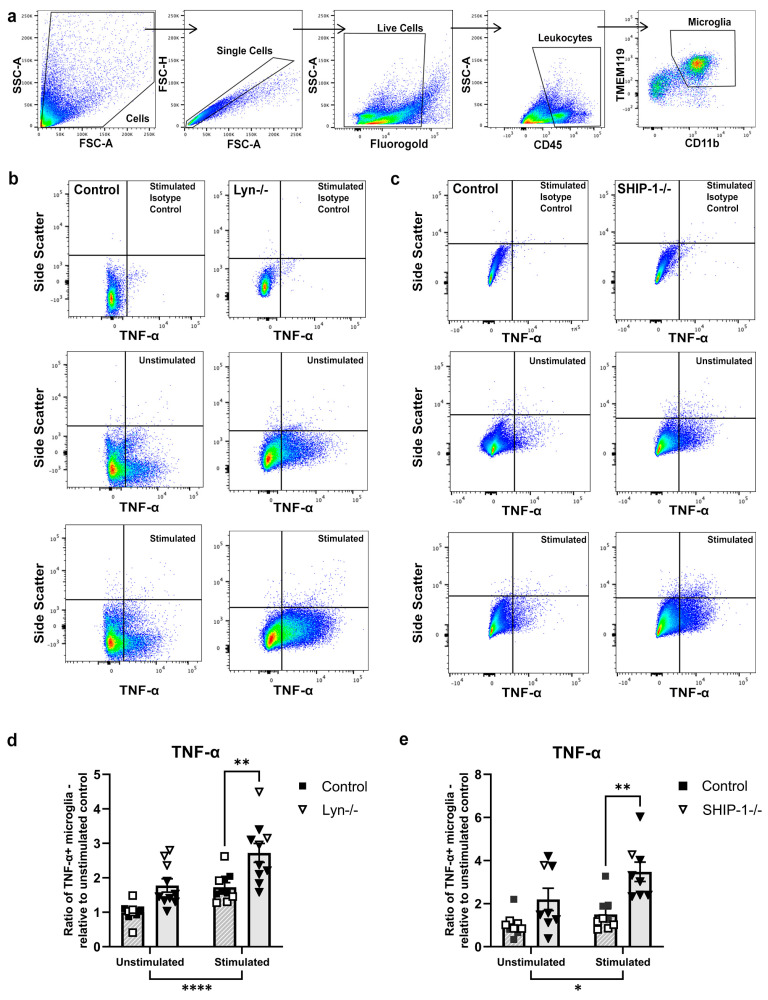
Microglia from Lyn−/− and SHIP-1−/− mice are hyper-responsive to LPS stimulation. (**a**) Gating strategy to select microglial populations from single-cell suspensions of brain. (**b**,**c**) Representative plots of the number of TNF-α+ microglia in stimulated isotype controls, unstimulated and stimulated brains from Lyn−/− and SHIP-1−/− mice. The ratio of TNF-α+ microglia from (**d**) Lyn−/− brains and (**e**) SHIP-1−/− brains. n = 8–10/group for flow cytometry analysis. Females = solid, males = open. Two-way ANOVA. Tukey post hoc test. * *p* < 0.05, ** *p* < 0.005, **** *p* < 0.0001.

## Data Availability

The datasets used and/or analyzed during the current study are available from the corresponding author on reasonable request.

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
