# Peer review of "Regulation of Microglial Signaling by Lyn and SHIP-1 in the Steady-State Adult Mouse Brain"

_cells, 2023, doi:10.3390/cells12192378_

Round 1

Reviewer 1 Report

The manuscript of Chu and co-workers "Regulation of Microglial Signaling by Lyn and SHIP-1 in the 2 Steady-State Adult Mouse Brain" decsribes the phenotypical analyses of both Lyn -/- and SHIP-/- mice with respect to their neuroinflammatory activity. They analyse both knockout mice in behavioural tests and use histochemical and FACS analyses to assess the microglial phenotype. This study is sound and comprehensive and well documented.

Minor point: One interesting finding of this study is a marked difference in spleen weight in SHIP-/- mice which is not seen in Lyn-/- mice. However, there is no explanation what causes these differences. I would suggest to perform a FACS analysis of spleen cells. 

Author Response

Minor point: One interesting finding of this study is a marked difference in spleen weight in SHIP-/- mice which is not seen in Lyn-/- mice. However, there is no explanation what causes these differences. I would suggest to perform a FACS analysis of spleen cells. 

Splenomegaly in young adult SHIP-1-deficient mice and its absence in 12-week-old Lyn deficient mice is expected based on previous reports. Both Lyn-deficient and SHIP-1-deficient mice have immune system perturbations with changes in the immune cell composition of spleen and this has been extensively documented in literature. Lyn-deficient mice have a loss of mature splenic B lymphocytes at all ages, and an expansion of plasma cells, nucleated erythroid cells and myeloid cells in spleen, which is only mild in 12-week-old mice and doesn’t lead to a change in spleen weight at this age, which only occurs as the mice age and develop lupus manifestations (reference # 48 in manuscript). In contrast, SHIP-1-deficient mice exhibit splenomegaly at 12-weeks of age, which is driven by a marked expansion of myeloid cells and nucleated erythroid cells as well as plasma cells, together with a loss of mature B cells (reference #12 in manuscript).

To address the Reviewer’s comment, two additional sections have been added to the manuscript as follows:

On page 9 (lines 266-269) in the first section of the Results on Lyn-deficient mice the following has been included: “While it is well-known that aged Lyn-deficient mice develop splenomegaly due to an expansion of myeloid cells as the mice develop lupus manifestations, young adult mice have a normal spleen weight albeit with an altered immune cell composition [48].”

On page 12 (lines 350-353) where the first studies on SHIP-1-deficient mice are presented, the following text has been included: “The early-onset splenomegaly phenotype of SHIP-1-deficient mice is driven by a marked expansion of myeloid cells and nucleated erythroid cells as well as plasma cells, together with a loss of mature B cells [12].”

Reviewer 2 Report

This submitted manuscript has employed two different KO mouse strains to explore the effects of Lyn and SHIP-1 on neuroinflammation and microglia activity, respectively, in vivo in steady-state.  The authors also investigate their effects on memory and anxiety related behaviors.  Overall, this is a descriptive but extensive manuscript utilizing different approaches to reveal microglia properties in vivo.  The manuscript was in good structure, logic writing, and excellent discussions.  The authors also pointed out their limitations on their research reflecting their deep understanding on this project.    

Suggestion:  the authors pointed out the microglia from these two strains showed exaggerated immune responses under LPS stimulation.  This phenomenon matches the theory: two-hit hypotheses, i.e., microglia were primed by the first stimulus and revealed exaggerated activation on the following stimulation.  It is better that the authors could discuss a little more on this direction.  Another minor suggestion is that the author should also include the data on il1b, il6 and ccl2 in addition to tnfα in the experiment LPS stimulation (Fig. 9).     

Author Response

Suggestion:  the authors pointed out the microglia from these two strains showed exaggerated immune responses under LPS stimulation.  This phenomenon matches the theory: two-hit hypotheses, i.e., microglia were primed by the first stimulus and revealed exaggerated activation on the following stimulation.  It is better that the authors could discuss a little more on this direction.  Another minor suggestion is that the author should also include the data on il1b, il6 and ccl2 in addition to tnfα in the experiment LPS stimulation (Fig. 9).  

To address the Reviewer’s first comment, we have added the following text to the discussion on lines 626 to 628: “Furthermore, in combination with microglial populations exhibiting an increased activated phenotype at baseline, SHIP-1 deficiency may promote a cumulative effect in microglial responses, similar to the “two-hit” hypothesis [see 84]”.

Regarding the Reviewer’s second comment, in the ex vivo microglial stimulation studies presented in Figure 9, we only assessed production of TNFα, as microglia are known to rapidly produce large quantities of this proinflammatory cytokine in response to LPS (refer to PMID: 30087595). We did not assess the other cytokine/chemokines suggested by the reviewer and do not have this additional data to include in the manuscript.

Reviewer 3 Report

To investigate the roles of Lyn and SHIP-1 in the steady-state of mouse central nervous system the authors used both male and female Lyn -/- & SHIP-1 -/- animals. Through brain structure analysis, behavior assessment, brain microglial measurement and gene expression and inflammation challenge, they concluded that the regulatory role of SHIP-1 in the CNS is prominent in the homeostatic state.

Comments: 

1) What is the male and female ratio (roughly) in the litters of Lyn-/- & SHIP-1-/- mice?

2) If using only male or female Lyn-/- & SHIP-1-/- mice in the study, may it generate distinct differences among the parameters measured?

3) Figure 2 Legend, line 301,  (g) soma area …   should be (h) soma area …

4)  Figure 4 Legend is missing description of (i-k).

5) Fig. 2g vs. Fig. 6g,  the Y-axis scale of Avg. branch length (mm) is drastically different – 10 fold more in Fig. 6g

Author Response

1) What is the male and female ratio (roughly) in the litters of Lyn-/- & SHIP-1-/- mice?

The distribution of the sexes of mice used in the study is approximately 60% males and 40% females for SHIP-1-/- and Lyn-/- mice. In all figures, the data of individual mice is shown and these have been separated into males (open symbols) and females (solid symbols).

2) If using only male or female Lyn-/- & SHIP-1-/- mice in the study, may it generate distinct differences among the parameters measured?

Due to breeding variability we combined both sexes for all of our measurements as for the most part, there were not enough numbers of each sex for statistical power. However, for some measurements like behavior, body weight and spleen weight, where numbers were greater, we have now conduced additional two-way ANOVA tests to examine the potential effect of sex. As such, the methods section from lines 250 to 254 now includes the following: “A two-way ANOVA test was also conducted to examine the factors of genotype and sex in behavior, body weight and spleen weight. However, examination of potential differences between sexes in other measurements were not statistically analyzed due to the lower numbers of animals of each sex and the unavoidable uneven distribution of sexes between experimental groups.”

In the results section on lines 269 to 271, the following text was added to reflect sex differences in body weight that was revealed by this analysis: “Post-hoc analysis revealed that body weight was modestly reduced in young adult female Lyn-/- mice compared to male Lyn-/- mice, which almost reached statistical significance (post-hoc analysis, p = 0.0505).”

3) Figure 2 Legend, line 301, (g) soma area …   should be (h) soma area …

We have made the change at line 460 of figure 2 legend from (g) soma area to (h) soma area to the figure legend.

4) Figure 4 Legend is missing description of (i-k).

Figure 4 legend has been changed from (a-h) to “(a-k) Gene expression analysis by qPCR of brain tissue from Lyn-/- and control mice. Gene expression is relative to the average of the control group.”

5) Fig. 2g vs. Fig. 6g, the Y-axis scale of Avg. branch length (mm) is drastically different – 10-fold more in Fig. 6g

Thank you for picking up this inconsistency. We have altered the Y-axis of figure 6g as that was an error, and have replaced the figure.